# Carbon Quantum Dots Prepared with Chitosan for Synthesis of CQDs/AuNPs for Iodine Ions Detection

**DOI:** 10.3390/nano8121043

**Published:** 2018-12-13

**Authors:** Juanjuan Song, Li Zhao, Yesheng Wang, Yun Xue, Yujia Deng, Xihui Zhao, Qun Li

**Affiliations:** 1School of Chemistry and Chemical Engineering, Qingdao University, Qingdao 266071, China; songjuan2017@163.com (J.S.); 2017020829@qdu.edu.cn (L.Z.); 2016020405@qdu.edu.cn (Y.W.); xueyun@qdu.edu.cn (Y.X.); dengyujia@qdu.edu.cn (Y.D.); qunli@qdu.edu.cn (Q.L.); 2State Key Laboratory of Bio-Fibers and Eco-Textiles, Shandong Collaborative Innovation Center of Marine Biobased Fibers and Ecological Textiles, Institute of Marine Biobased Materials, Qingdao University, Qingdao 266071, China

**Keywords:** carbon quantum dots, chitosan, gold nanoparticles, Iodine ions detection

## Abstract

Water-soluble and reductive carbon quantum dots (CQDs) were fabricated by the hydrothermal carbonization of chitosan. Acting as a reducing agent and stabilizer, the as-prepared CQDs were further used to synthesize gold nanoparticles (AuNPs). This synthetic process was carried out in aqueous solution, which was absolutely “green”. Furthermore, the CQDs/AuNPs composite was used to detect iodine ions by the colorimetric method. A color change from pink to colorless was observed with the constant addition of I^−^ ions, accompanied by a decrease in the absorbance of the CQDs/AuNPs composite. According to the absorbance change, a favorable linear relationship was obtained between ΔA and I^−^ concentration in the range of 20–140 μM and 140–400 μM. The detection limit of iodide ions, depending on the 3δ/slope, was estimated to be 2.3 μM, indicating high sensitivity to the determination of iodide. More importantly, it also showed good selectivity toward I^−^ over other anion ions, and was used for the analysis of salt samples. Moreover, TEM results indicated that I^−^ ions induced the aggregation of CQDs/AuNPs, resulting in changes in color and absorbance.

## 1. Introduction

Iodine is one of the essential trace elements in the human body which plays a significant role in maintaining the function and morphology of the thyroid [1,2]. The World Health Organization determines that iodine consumption per day for adults and during lactation or pregnancy is 150 μg or 250 μg, respectively [3]. Abnormal concentrations can cause serious harm to the human body. Iodine deficiency at critical stages of pregnancy and early childhood can lead to impaired brain development, leading to impaired mental function [4,5]. Iodized salt is the main method used to prevent iodine deficiency disorders. However, excessive intake of iodine may cause hyperthyroidism [6]. These are common health problems, especially in developing countries [7]. Therefore, developing a simple and convenient method for detecting iodide ions has important significance for iodine supplementation. There are methods such as voltammetry [8,9], spectrofluorimetry [10,11,12], spectrophotometry [13], potentiometry [14] and chromatography [15,16], among others. However, these methods have some limitations, including how time-consuming they are and the use of tedious procedures and sophisticated instrumentations, which limit their application. Therefore, it is very desirable to develop a simple and fast method for the detection of iodine ions with selectivity and sensitivity. Fortunately, colorimetric assays based on metal nanoparticles, especially, gold nanoparticles (AuNPs) and silver nanoparticles (AgNPs), can provide a simple way to solve these limitations. Additionally, compared with the well-developed chromophoric chemosensors and carbon quantum dot fluorophore-based fluorescence methods [17], colorimetric assays based on AuNPs and AgNPs are of particular interest as they possess high extinction coefficients in the visible region, and the colors of the dispersed and aggregated nanoparticle solutions are different. The detection results can be clearly measured with a simple spectrophotometer or even by the naked eye [18,19].

AuNPs have attracted great attention recently due to their special physical and chemical properties caused by local surface plasmon resonance (LSPR), so they have been used in many fields, including catalysis [20,21], chemical and biological sensors [22,23,24] and surface-enhanced Raman scattering [25,26]. AuNPs are usually synthesized by using a variety of reducing agents in aqueous solution, and the most popular method still used today was devised by Turkevich et al. [27]. Now, more and more materials have been developed to synthesize gold nanoparticles. Among them, carbon-based nanomaterials are very interesting and have gained tremendous attention in the past few years. Of note are carbon quantum dots (CQDs), which are carbon nanomaterials with sizes less than 10 nm. CQDs have been reported recently as reducing agents for the synthesis of metal nanoparticles for sensing applications. Liu et al. reported Ag/CQDs composites prepared with CQDs as both a reducing agent and stabilizer for Hg^2+^ ion detection [19]. Niu et al. developed a “turn on” fluorescent sensor for Pb^2+^ detection based on Grapheme quantum dots (GQDs) and AuNPs [28]. Moreover, Amjadi et al. reported glucose-derived CQDs as a reducing agent and stabilizer for the synthesis of CQD/Ag nanocomposites and exploited them for the colorimetric detection of methimazole [29]. Although these methods have various advantages, the synthesis procedures in most of them require high temperatures and/or long reaction times.

Chitosan is a linear polysaccharide containing –OH and –NH_2_ groups. As the second most plentiful natural biopolymer, it is nontoxic, biocompatible, biodegradable and relatively inexpensive. However, chitosan has been discarded as seafood waste for a long time. In this work, water-soluble and reductive CQDs were synthesized via the hydrothermal method with chitosan as the carbon source. The CQDs, prepared with chitosan as both a reducing agent and stabilizer, were further used for the synthesis of AuNPs. The whole synthetic process is environmentally friendly. Furthermore, a simple, rapid, selective and sensitive colorimetric method, based on CQDs/AuNPs composite, was fabricated for the detection of iodine ions, indicating potential application in foodstuffs analysis in the future.

## 2. Materials and Methods

### 2.1. Materials

Chitosan (deacetylation degree 82.5%) was prepared in our lab with shrimp shell, which is usually discarded as shellfish waste in the fishing and seafood industries. The deacetylation degree was determined by titration. The detailed test is supplied in the Appendix A. HAuCl_4_·4H_2_O and other chemical reagents (analytical grade) were purchased from the Beijing Chemical Factory (Beijing, China) and all the reagents were used as-received, without further purification. Ultra-pure water (Millipore, ElPaso, TX, USA) was used for all the experiments.

### 2.2. Instruments

Transmission electron microscopy (TEM) images were obtained by a JSM-2100Plus (JEOL, Tokyo, Japan) microscope at an accelerating voltage of 100.0 kV, equipped with an energy-dispersive spectroscopy (EDS) detector. The UV–vis absorption spectra were recorded with a Shimadzu UV3150 spectrophotometer (Kyoto, Japan). Fluorescence spectra were measured with a Fluoromax–4 Spectrofluorometer (HORIBA, Kyoto, Japan).

### 2.3. Synthesis of CQDs

The CQDs were prepared by the hydrothermal method using chitosan solution as a reaction precursor according to the method reported by Yang [30] with some modification. The typical synthesis is described as follows: firstly, 0.5 g chitosan was added to 100 mL of 1% acetic acid solution and dissolved at room temperature using ultrasonic technique, and then the prepared chitosan solution was filtered with an aperture of 0.45 μm to remove the insoluble substance. Secondly, 20 mL of 0.5% chitosan solution was placed in an autoclave (50 mL) and heated at 180 °C for 12 h. Thirdly, when cooled down to room temperature, the brownish black solution was filtered by a membrane with an aperture of 0.22 μm. Next the solution was centrifuged at 10000 rpm for 15 min to remove all deposits and yield a canary yellow CQD aqueous solution, and then the as-prepared CQDs were stored in a refrigerator at 4 °C for future use. The final CQDs were denoted as CHI-x, where x referred to the carbonization time.

### 2.4. Synthesis of CQDs/AuNPs

CQDs/AuNPs were synthesized by the reduction of HAuCl_4_ with CQDs as a reducing agent. Firstly, the HAuCl_4_ (24 mM, 75 μL) and CQD solutions (diluted five times 3 mL) were added into a 5 mL reaction bottle. Then, the solution was vigorously stirred for 5 min, during which the solution changed from being a colorless liquid to a pink liquid. Finally, the resulting solution was reacted at room temperature for 3 min. The obtained solution was a purplish-red color, demonstrating the formation of CQDs/AuNPs.

### 2.5. Detection of I^−^

For the detection of I^−^, 200 μL of CQDs/AuNPs was added into a centrifuge tube and then 8 μL of KI solution was added to the tube. There was an observed color change in the sample after incubation for 5 min at room temperature. In addition, 3 mL of CQDs/AuNPs was added into a cuvette of ultraviolet spectrophotometer. Then, the potassium iodide (10 mM) was added into the cuvette 20 times, with a concentration gradient of 6 μL, under gentle shaking. The absorption spectra were recorded in the range of 300–800 nm.

To evaluate the selectivity, 400 μL of anion solutions including F^−^, Cl^−^, Br^−^, H_2_PO_4_^−^, HPO_4_^2−^, PO_4_^3−^, S_2_O_8_^2−^, SO_4_^2−^, SO_3_^2−^, HCO_3_^−^, CO_3_^2−^ and NO_2_^−^, were respectively mixed with 3 mL of CQDs/AuNPs. Photographs and UV–vis spectra were taken after I^−^ ions were added to the CQDs/AuNPs for 5 min.

## 3. Results and Discussion

### 3.1. Characterization of CQDs

In this work, highly fluorescent CQDs were fabricated by hydrothermal carbonization of chitosan at 180 °C. In order to explore the best reaction conditions for the synthesis of CQDs, we changed the reaction time of carbonization. As elaborated in Figure 1B,C, after six hours of carbonization, the chitosan solution turned light yellow in color, while the solution displayed a bright blue light under the UV light lamp with excitation at 365 nm until carbonized for eight hours. CHI-12 and CHI-24 have almost the same fluorescence intensity, so we chose CQDs (CHI-12) as a reductant and stabilizer for the synthesis of gold nanoparticles.

The optical properties of CQDs (CHI-12) were characterized by UV-Vis and fluorescence (FL) spectroscopies. As shown in Figure 1A, the CQDs have a broad absorption band at 285 nm which is attributed to the n–π* transition of C=O [29], while chitosan does not have any absorption bands above 220 nm. The fluorescence spectra of CQDs are shown in Appendix A. It can be seen that when excited at a maximum excitation wavelength of 330 nm, the prepared CQDs exhibit narrow and symmetrical fluorescence spectra curves at a maximum emission wavelength of about 405 nm. Moreover, as the carbonization time increases, the fluorescence intensity of CQDs gradually becomes stronger. Additionally, the fluorescence emission spectra of CQDs were further investigated under different excitation wavelengths ranging from 280 nm to 480 nm with 20 nm intervals (Appendix A). With the increase of the excitation wavelength from 280 nm to 480 nm, the FL intensities decrease remarkably. Moreover, in normalized fluorescence emission spectra, the excitation-dependent emission (EDE) phenomenon could be observed. The maximum emission peak shifted from 394 nm to 503 nm with an increase of the excitation wavelength from 280 nm to 480 nm. The red shift phenomena may be caused by the different sizes and multi-surface emission sites of CQDs [31]. Additionally, some works on carbon dots also suggested that the optical properties of carbon dots are mostly determined by fluorescent surface groups [32,33].

The structures and morphologies of the as-prepared CQDs were investigated by transmission electron microscopy (TEM) and the image is presented in Figure 1D,E. The TEM image showed that the as-prepared CQDs were well-dispersed with average diameters of about 2 nm. The high-resolution TEM (HRTEM) image shows that the CQDs have obvious lattices, and one nanoparticle reveals a lattice spacing of 0.212 nm, which corresponds to the (100) lattice space of graphitic carbon.

### 3.2. Characterization of CQDs/AuNPs

The surface of the synthesized CQDs has many functional groups, which can improve its water solubility and reducing ability. The aqueous solution of the diluted carbon quantum dots is almost colorless, but after the carbon quantum dots react with Chloroauric acid, the solution change to purplish red (Figure 2A inset). The UV-vis absorption spectrum of the products (Figure 2A) has an absorption peak at 537 nm, which is the typical plasmon resonance absorption peak of AuNPs, indicating that AuNPs have been synthesized. Additionally, the absorption peak at 285 nm decreased slightly, indicating that the carbon quantum dots were in an excessive state. Moreover, compared with the carbon quantum dots, the fluorescence of CQDs/AuNPs was obviously weakened. The morphology and size of the CQDs/AuNPs were characterized by TEM (Figure 2B), where spherical particles of CQDs/AuNPs with uniform distribution and particle size of about 20–30 nm can be observed. The size distribution histogram of CQDs/AuNPs is shown in Appendix A. The phase and formation of the CQDs/AuNPs was further confirmed by HRTEM. The HRTEM of CQDs/AuNPs (Figure 2C) reveals lattice fringes with inter-planar spacing of 0.238, which are attributed to the (111) planes for face-centered-cubic (f c c) gold. As illustrated in Figure 2D, it is noteworthy that CQDs surrounded the AuNPs obviously. This indicates that CQDs are not only responsible for enriching the Au^3+^ ions to promote the reduction reaction, but also play a role as protective coatings to prevent aggregation of the gold nanoparticles [34,35]. Moreover, the particle size of CQDs/AuNPs is clearly larger than that of carbon quantum dots. The corresponding selected area electron-diffraction (SAED) pattern (Appendix A) revealed that CQDs/AuNPs composites are poly crystal and confirmed that the C element is contained in the nanoparticles and these C elements should be derived from the CQDs. Furthermore, the energy dispersive X-ray spectroscopy (EDX) diagram of single nanoparticles (Appendix A) shows that the spectrum consisted of different peaks from C, O, Au and Cu. The Cu and part of the C peaks can be attributed to the C-coated Cu TEM grid used for the analysis [36]. The O and other part of C peaks correspond to CQDs used for the synthesis of AuNPs. The Au peak came from the AuNPs. The above Uv-vis, TEM, SAED and EDX results confirmed that gold nanoparticles were synthesized by using CQDs as a reducing agent.

### 3.3. Optimization of Experimental Conditions

In order to explore the optimal conditions for the synthesis of gold nanoparticles, we changed the conditions for the synthesis of gold nanoparticles. So, we synthesized AuNPs under different concentrations of HAuCl_4_ and different dilution multiples of CQDs. As we can see from Figure 3A, when the concentration of chloroauric acid is constant, the absorption peak increased with the increase of the concentration of CQDs. Meanwhile, it is worth noting that the reaction time presents as a monotonous decrease with increasing the concentration of CQDs. It can be seen from Figure 3B that the larger the concentration of chloroauric acid, the higher the absorption peak of the gold nanoparticles, and the deeper the color of the samples (Figure 3C). Meanwhile, the reaction time increased with the increase of the concentration of chloroauric acid. However, the gold nanoparticles will agglomerate with the increasing concentration of chloroauric acid, so we selected 0.6 mM of chloroauric acid, diluted five times by the carbon quantum dots to synthesize gold nanoparticles. The reaction time is 5 min.

### 3.4. Sensitivity of the Sensing System

After searching for the optimal synthesis conditions of the gold nanoparticles, gold nanoparticles were synthesized and iodine ions were detected. The absorption spectra of CQDs/AuNPs were measured for quantification of iodine ions. The color changes and the aggregation state changes of CQDs/AuNPs could be observed by the naked eye with different concentrations of iodine ions. From Figure 4A, we can find that with the increasing concentration of iodine, the absorption peak of gold nanoparticles at 537 nm decreased, and that the peak at 750 nm increased, and the color of CQDs/AuNPs changed gradually from purple-red to a light red (Figure 4B). However, as the concentration of iodide ions continued to be increased, the absorbance at 537 nm and 750 nm decreased, and the color of the gold nanoparticles changed from light red to colorless. When the concentration of iodide increased to 400 μM, the color of the gold nanoparticles was faded, indicating that the gold nanoparticles reacted with the iodine ions. Therefore, CQDs/AuNPs can be used to detect iodine ions by the colorimetric method. When the concentration of iodine is 200 μM, the color change of the gold nanoparticles is obvious. Furthermore, Figure 4C shows that there are favorable linear correlations between ΔA and I^−^ concentration in the range of 20–140 μM and 140–400 μM. When the concentration in the range of 20 to 140 μM, k_1_ = 0.58, R_1_^2^ = 0.96, and when in the range of 140 to 400 μM, k_2_ = 1.61, R_2_^2^ = 0.99. According to the linear relationship between the concentration of I^−^ and ΔA, the detection limit of the iodine ions relied on the 3δ/slope and was estimated to be 2.3 μM. The above results indicated that this method possessed good linearity and high sensitivity for the determination of iodide. By comparing the detection parameters of iodide ions (as shown in Appendix A), our sensor is a new sensor with comparable sensitivity.

### 3.5. Selectivity of the Sensing System

To evaluate the selectivity of this sensing system, twelve other anion ions, namely F^−^, Cl^−^, Br^−^, H_2_PO_4_^−^, HPO_4_^2−^, PO_4_^3−^, S_2_O_8_^2−^, SO_4_^2−^, SO_3_^2−^, HCO_3_^−^, CO_3_^2−^ and NO_2_^−^, were selected to investigate the selectivity of this dual-signal sensor for the detection of I^−^ under the optimal conditions. The concentrations of all ions were 400 μM. It could be observed from Figure 5A that the absorption spectra of CQDs/AuNPs composite solutions after the addition of I^−^ decreased significantly, whereas the absorption peak hardly had any change when other anion ions were added to the CQDs/AuNPs solution. Although NO_2_^−^ caused absorption blue-shift, the decrease was still not obvious. CQDs/AuNPs displayed a purple-red color after adding other anion ions, but the I^−^ made the system change color from purple-red to colorless (Figure 5C). At the same time, from Figure 5B, it is clear that when other anion ions were added to the CQDs/AuNPs solution, the change in ΔA was very weak. These results demonstrate that CQDs/AuNPs possess high selectivity toward I^−^ detection.

Additionally, in order to evaluate the feasibility of this method, the proposed detection method was applied for I^−^ detection in salt samples. The result is shown in Appendix A; good recoveries between 98.4% and 107.4% were obtained, suggesting that the colorimetric method based on CQDs/AuNPs may be of great potential for I^−^ detection in real samples.

### 3.6. Mechanism for CQDs, CQDs/AuNPs Formation and Iodide Ion Detection

Scheme 1 proposed the schematic illustration of preparing CQDs and CQDs/AuNPs, and detection of I^−^ based on CQDs/AuNPs. The CQDs were firstly synthesized by the hydrothermal carbonization of chitosan at 180 °C. At the same time, chitosan chains were broken under the high temperature, yielding amino and hydroxyl groups. Broken chains were further carbonized into CQDs and the CQDs will be functionalized by the functional groups on its surface, which can improve its water solubility and reducing ability. Then, the CQD/AuNP composite was further prepared with the functional CQDs as the reducing agent. The characteristic feature of this synthetic process is that it occurs in aqueous solution, and has the advantage of being very cheap and absolutely “green”. Furthermore, the CQDs/AuNPs composite was used to detect iodine ions by the colorimetric method. The addition of I^−^ ions results in a gradual color change of CQDs/AuNPs from purple-red to light red and even colorless. The morphological change of the CQDs/AuNPs was investigated by TEM. From TEM images (Figure 6), we could see that the addition of I^−^ ions induced the aggregation of AuNPs. However, the aggregation of AuNPs did not occur in the presence of any other anion ions. Therefore, it could be confirmed that the change of the solution color and the absorption intensity in Figure 3C were closely associated with the aggregation of CQDs/AuNPs.

## 4. Conclusions

In summary, the functional CQDs with average diameters of about 2 nm were synthesized firstly by the hydrothermal carbonization of chitosan. Then, AuNPs were further synthesized with the prepared CQDs acting as a reducing agent and stabilizer, and the particle size was about 20–30 nm. Furthermore, the CQDs/AuNPs composite was used to detect iodine ions by the colorimetric method. Color change from pink to colorless and a decrease in the absorbance of the CQDs/AuNPs composite could be measured with an increasing amount of I^-^ ions. Additionally, a favorable linear relationship was obtained between ΔA and I^−^ concentration in the range of 20–140 μM and 140–400 μM. The detection limit of iodide ions, depending on the 3δ/slope, was estimated to be 2.3 μM. More importantly, it also showed good selectivity toward I^−^ over other anion ions, and has been successfully applied for the analysis of salt samples. Furthermore, the TEM results indicated that I^−^ ions induced the aggregation of CQDs/AuNPs, resulting in changes in color and absorbance.

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
