# Peer review of "Carbon Quantum Dots Prepared with Chitosan for Synthesis of CQDs/AuNPs for Iodine Ions Detection"

_nanomaterials, 2018, doi:10.3390/nano8121043_

Reviewer 1 Report

The manuscript describes the fluorescence detection of iodine ions based on carbon quantum dots prepared from chitosan and decorated with Au nanoparticles. Nanomaterials used are well characterized and promote for sensitive analyte detection within micromolar range of concentrations. Interferences that could affect measurement results are correctly specified and show minimal influence on iodine signal. The manuscript can be recommended for publication after minor changes:

1. Technical notes. line 42: field instead of filed, line 83 – WERE synthesized, Figure 1(A) – absorbance is dimensionless, remove ‘(nm)’ from ordinate legend or add (a.u.) as in Fig. 2(A), Fig. 3(A, B), Figure 2(C) – the numbers should be increased, line 203 – the number of 1. significant decimals should be reduced to two. Concentration range cannot start with zero concentration – give first (minimal) concentration tested.

2. Instrumentation used should be described in Materials and Methods section, especially those of SEM/TEM experiments

3. Metrological assessment of the sensing particles properties is necessary together with comparison of analytical performance with analogs described in the literature.

Author Response

Manuscript ID: Nanomaterials-403845

Title: Carbon quantum dots prepared with chitosan for synthesis of CQDs/AuNPs for Iodine ions detection

Journal: Nanomaterials

Correspondence Author: Xihui Zhao

Dear reviewer,

Thank you for your kind letter on my article (MS Number: Nanomaterials-403845)! We also want to express our deep thanks to your positive comments. Those comments are all valuable and very helpful for revising and improving our paper, as well as the important guiding significance to our research. We have revised the manuscript according to your kind advice and detailed suggestions. Enclosed please find the responses to the comments. Thank you very much for all your help and we are looking forward to hearing from you. If you have any question about this paper, please don’t hesitate to let us know.

Sincerely yours,

Xihui Zhao

E-mail address: [email protected]

Reviewer 2 Report

In the current manuscript, the authors show the results of synthesis of the functional carbon dots with average diameters of about 2 nm. They report of subsequent synthesis of gold nanoparticles with the prepared carbon dots as reducing agent and stabilizer. CDs/AuNPs composite was used to detect iodine ions by colorimetric method. Color change from pink to colorless and decrease in the absorbance of the CDs/AuNPs composite was measured with increasing amount of I- ions. Using TEM imaging, it was shown that I- ions induced the aggregation of CQDs/AuNPs, resulting in the color and absorbance change. The presentation of the data is in general clear; however, certain grammatical correction of the text is needed. I believe that the current work is of potential interest for specialists working in this field. Therefore, I recommend it for publication after addressing several issues listed below.

First, I believe that the mane carbon QUANTUM dots is misleading, since it makes an impression that the photo-physical properties of the particles are determined by quantum confined exciton in the particles’ core. Whereas, in some of early works on carbon dots it was suggested that exciton can be generated in carbonic core of the particles, further studies showed that their optical properties are mostly determined by fluorescent surface groups, see for instance Nano Lett., 2014, 14, 5656;J. Phys. Chem. Lett., 2017, 8, 5751, and many others. Successful use of carbon dots for sensing application also confirms this.

Figures that show absorption spectra do not need any units for the vertical scale. Absorbance is the logarithm of a ratio of intensities and hence does not have any units.

In figure S1, the authors have forgotten to add either counts or a.u. to the vertical axis label.

Labeling of sub-images in figure 1 is confusing. Image E contains parts that are also labeled as a,b and c.

Some of the labels in figures are hardly readable. For instance, Figure 1E(b) and (c): the numbers are too small and unreadable.

Author Response

(The authors gave the same response as above.)

Reviewer 3 Report

As regards the novelty:

- Line 49: The original idea of the work, understood as the determination of iodide ions via quenching of the fluorescence of CQD fluorophores is based on the article "Determination of Iodide via Direct Fluorescence Quenching at Nitrogen-Doped Carbon Quantum Dot Fluorophores" (Environ. Sci. Technol. Lett., 2014, 1 (1), pp 87-91; DOI: 10.1021/ez400137j). The authors have limited their study to the substitution of N-doped CQDs for CQDs+AuNPs, an idea that is not original either, in view of the existence of such composites in the bibliography. The introduction should thus clarify this points.
- Line 53: The existence of these studies also limits the novelty of the work presented in this paper.

Regarding the applicability:

- Lines 34-35 ("[...] detecting iodide ion has important clinical significance."): The measurement of iodide ion has no clinical significance. In Clinical Chemistry the parameters of interest are Triiodothyronine (total and free) and Tetraiodothyronine (total and free). There are methods of determination, (especially electrochemiluminescence) highly optimized in terms of speed, accuracy and precision. This must be clarified in the paper, and -consequently- the applicability should not be related with Clinical Chemistry, so the authors must suggest other possibilities.

Other issues:

- This paper would benefit from some closer proofreading. It includes some linguistic errors (e.g. agreement of verbs) that at times make it difficult to follow. For instance, in the abstract: "[...] As a reducing agent [...]" (indefinite article missing); "[...] and was absolutely green [...]" (verb missing); "[...] Furthermore, the CQDs/AuNPs composite [...]" (definite article missing); "I-" (superscript); "[...] and was used for the analysis [...]" (verb agreement).

Author Response

Manuscript ID: Nanomaterials-403845

Title: Carbon quantum dots prepared with chitosan for synthesis of CQDs/AuNPs for Iodine ions detection

Journal: Nanomaterials

Correspondence Author: Xihui Zhao

Dear reviewer,

Thank you for your kind letter on my article (MS Number: Nanomaterials-403845)! We also want to express our deep thanks to your positive comments. Those comments are all valuable and very helpful for revising and improving our paper, as well as the important guiding significance to our research. We have revised the manuscript according to your kind advice and detailed suggestions. Enclosed please find the responses to the comments. Thank you very much for all your help and we are looking forward to hearing from you. If you have any question about this paper, please don’t hesitate to let us know.

Sincerely yours,

Xihui Zhao

E-mail address: [email protected]

Round  2

Reviewer 3 Report

The issues raised in previous iteration have been addressed. As far as I am concerned, no further changes are required.